# Economic Integrated Structural Framing for BIM-Based Prefabricated Mechanical, Electrical, and Plumbing Racks

Seungchan Baek [1], Jongsung Won [2],* and Sejun Jang [3],*

1 Department of Architecture, Kyungil University, Gyeongsan 38428, Republic of Korea
2 Department of Architecture, Korea National University of Transportation, Chungju-si 27469, Republic of Korea
3 School of Architectural Engineering, Kunsan National University, Gunsan 54150, Republic of Korea
* Correspondence: jwon@ut.ac.kr (J.W.); jang@kunsan.ac.kr (S.J.); Tel.: +82-063-469-4785 (S.J.)

**Abstract:** Prefabrication, one of the methods to increase productivity by moving construction activity to factory work, has evolved into multi-trade prefabrication. Although previous studies have introduced the merits and directions of multi-trade prefabrication technology, various design and installation techniques must be developed for the economical application of multi-trade prefabrication. This study aims to provide an economical design for prefabricated mechanical, electrical, and plumbing (MEP) rack (PMR) structural framing. We proposed five types of PMR structural framing techniques and analyzed their required channel material and labor inputs through a pilot test. The efficiency of PMR structural frames was examined by adjusting the supporting interval, moving the duct outside, and removing the upper framing. Economics and productivity analysis revealed that removing the upper framing method was the most effective when the coordination period was secured. Adjusting the supporting intervals is also an economical design option. The findings of this study can help enhance the economic feasibility of prefabrication and modularization of construction and their widespread utilization.

**Keywords:** multi-trade; prefabrication; structural framing; modular construction

## 1. Introduction

In recent years, the application of various technologies, such as information technology, laser scanners, and modularization, has increased construction productivity. With the introduction of building information model (BIM) technology, problems in the design stage and conflicts in the construction stage are prevented in advance [1]. In addition to introducing BIM, which creates a virtual 3D world and detects faults, the method of collecting information in the real world has also been improved. A laser scanner is used to check whether the structure designed in the pre-construction stage is modified or matched with the drawing, and the structure can be constructed using its 3D model and BIM [2]. In addition, information about the geographic features of a wide area can be obtained rapidly using drones and photogrammetry technology [3]. Despite introducing these technologies, the actual environment in which a structure is constructed can be improved only when the production processes, such as prefabrication, are enhanced in parallel [4]. Prefabrication technology is concerned with the construction production process and serves as a manufacturing platform to improve productivity and safety management [5].

Prefabrication is not a new technology and is used for various purposes. It has been undergoing improvement and optimization for a long time [6]. Applying precast concrete elements to a concrete field can reduce the curing time for constructing structures in the field [7]. Using curtain walls with glasses and frames can reduce the assembly time in the field and avoid the use of scaffolding outside the building. It can eliminate various conflicts related to construction [8]. Using prefabrication technology can shorten the construction period because the components are manufactured in advance in a factory instead of at the

construction sites. In particular, it can produce structures repeatedly [9]. Furthermore, the learning effect of the labor units involved in repeated productions increases the efficiency of the work and productivity [10]. The conversion to factory production reduces the processing of materials in the field. The total waste generation rate of construction projects using prefabrication technology is 25.85% lower than that of non-assembled projects. Using prefabrication can effectively reduce most construction wastes (inorganic non-metallic, organic, metallic, and composite wastes) [11,12]. For this reason, many countries encourage the introduction of prefabrication or modular construction techniques, which is gradually increasing [13].

Recently, owing to the complexity of building systems, the scope of prefabrication has been expanded from construction works such as precast concrete and curtain walls to mechanical, electrical, and plumbing (MEP) elements [14]. In particular, a recent trend in buildings is the use of heavy MEP systems. The prefabrication of MEP components is becoming more sophisticated and advanced, owing to the introduction of BIM technology [15]. Advances in pre-design review technologies such as BIM have improved the environment where MEP coordination can be performed in advance. Therefore, single-trade prefabrication techniques have been replaced with multi-trade prefabrication techniques that combine the elements of complex works in advance [16,17]. The multi-trade prefabrication technology requires substantially more effort and time in designing and manufacturing than single-trade prefabrication because it solves conflicts between different work types in advance and manufactures by attaching elements of various work types within one integrated frame. However, it minimizes the fieldwork and reduces the construction period; hence, it holds advantages [4]. The benefits of combining various elements in one space can be expected to reduce air and increase productivity more than the existing single-trade prefabrication techniques [18].

A prefabricated MEP rack (PMR) is the most commonly used multi-trade prefabricated member. Many studies have demonstrated its advantages, such as reducing the construction period owing to the conversion of fieldwork into pre-factory work and productivity improvement [19,20]. Furthermore, it improves safety remarkably, as it can be used to manage quality improvement and flammable work in the factory [21]. However, despite its various advantages, the application of PMR increases cost. The prefabrication technology has been reported to have a limited application and needs technical improvement and government support for economic application [22,23]. Detailed discussions on various design and construction techniques for the economic application of PMRs are required.

One of the most widely studied economic factors driving the use of PMRs is the economic design of the structural frames [10]. This study sheds light on the economic structural framing of PMRs, which has recently become the most actively discussed multi-trade prefabrication technique. For PMR application, integrated structural framing for combining various members is required. Because of costs incurred in addition to the expenses for the existing MEP elements, the contribution of MEP corridor racks to economic growth may become significantly lesser than that of the conventional method. Studies on detailed design methods of PMRs are insufficient to date. This study proposed five methods for designing PMRs and suggested ways to measure their economic feasibility by comparing three aspects: design cost, material cost, and manufacturing cost. In addition, a structural framing technique for the economic application of PMRs was proposed.

This article is organized as follows. First, we investigate the existing literature on multi-trade prefabrication and analyze the ways of improving the application of PMRs. Second, the research method is presented, and the case study details are provided. Third, five types of PMR structural frames are proposed and described. We also present the results of the efficiency analysis experiments for these five types of frames. Finally, we discuss the type of structural framing that can be economical.

## 2. Literature Review on Prefabrication

Although the prefabrication technique has several advantages, including an increase in efficiency, various studies have revealed the disadvantages and limitations of prefabrication. If a design prepared in the design stage changes, the projects planned with prefabrication require considerable coordination effort and time, owing to the utilization of BIM, to prevent more re-work than that required with a conventional method [24]. Prefabrication requires more materials for manufacturing components than the conventional method and increases costs because the prefabricated members must maintain a stable shape during production, transportation, and construction [25]. Prior to the full-scale construction stage, transportation costs are incurred additionally because pre-produced construction members must be transported from the factory to the construction site [26]. In a prefabrication construction process, units heavier than the units used in the conventional method are lifted. Therefore, it increases the required equipment capacity, and expensive cranes should be used [27]. In addition, it requires a high level of project management technique. Lack of field space hinders the application of prefabrication technology. To solve this problem, project management techniques such as just-in-time management are required, and design changes must be prevented [23,28]. Furthermore, global positioning system and radio frequency identification, which are innovative technologies, are necessary for the production, transportation, and installation management of the prefabricated construction members [29].

For the above reasons, projects do not derive economic benefits from applying prefabrication unless certain conditions are satisfied. Many researchers have studied the framework for the economical use of prefabrication. According to Li et al. [29], directly considering the economic effects of prefabrication in the construction stage is difficult and must be approached from a national standpoint, such as waste treatment. Cost subsidies implemented by government policies have been analyzed to be a crucial way to promote prefabrication. Tam et al. [26] investigated the overall construction process in a prefabrication case study and suggested that the application of prefabrication made various changes, such as those in joints, lifting equipment, and finishing methods, and the economic feasibility depended on the construction method. Hong et al. [30] analyzed the cost variations caused by prefabrication and suggested material cost as the most significant contributor to the increase in cost. Jang [31] indicated that a major cause of the rise in cost due to the application of prefabrication was the construction cost of structural framing and units that fix the prefabricated materials in the manufacturing, transportation, and construction stages.

## 3. Research Method

In this study, the PMR is not a racking system supporting high loads but artificially integrates MEP elements supported by individual hangers. The construction drawings did not suggest a structural supporting system in which structural materials such as H beams are used but targeted MEP elements that collectively apply individual hangers. Pipes with a diameter of 100 mm or more, requiring a system channel of more than 50 mm, were excluded from the PMR integration to avoid lowering the ceiling height of the corridor. Although the MEP elements was bulky (similar to a duct), the elements that a 50 mm system channel could support were integrated into the PMR.

In summary, the PMR of this study (1) deals with MEP elements supported by hangers, (2) excludes MEP elements requiring structural reinforcement, (3) targets corridors, and (4) includes MEP elements that can be supported by 50 mm channels.

This study aims to identify an economical design for PMR structural frames among five designs. The data acquisition and comparison of economic feasibility for the five designs considered the following three factors:

- Design cost: person-day input for the design phase per cubic meter.
- Material cost: system channel input for manufacturing per cubic meter.
- Manufacturing cost: person-day input for manufacturing phase per cubic meter.

First, the basic designs of the PMR frames were prepared in the design stage, and the detailed coordination process for attachment was added for each type of frame. The increase in cost owing to this design process must be considered while applying the prefabrication method [18]. Second, the structural frame was assembled; the various construction members were combined and fixed in the form of a unit during transportation and construction. This step contributes the most to increasing the material cost. This study proposed PMR structural frame designs using lightweight system channels and calculated the cost of the system channel. Third, the labor input needed to produce the PMR units in the manufacturing stage was analyzed. The construction stage may be affected by the lifting equipment, construction method, and site conditions [29]. This study analyzed the material and labor costs required to produce PMR units up to the manufacturing stage.

Studies on the structural framing of PMR have not been specifically reported; however, structural framing is one of the factors increasing the cost of PMR compared with traditional hanging systems [31]. Engineers and manufacturers were consulted to propose economic structural framings. The study aimed to reduce structural framing used for modularization. We identified five types of PMR structural frames through preliminary verification of constructability and economic feasibility and conducted mock-up tests on the five types.

The five types of PMR modules were applied to different areas, and the input resources were calculated per cubic meter. This is because one cubic meter can be used to measure the density of an MEP system and the difficulty of coordinating the MEP system [32]. If the resources required for manufacturing PMR were calculated per square meter, calculating the effect would be difficult when the plenum space was low or high. Moreover, a PMR is a volumetric module that considers the modularization of space; hence, it was reasonable to calculate the amount of input resources per cubic meter. This study is based on a case study (Table 1).

**Table 1.** Case Study (partially supplemented).

|  | Type A | Type B | Type C | Type D | Type E |
|---|---|---|---|---|---|
| Length (m) | 90 | 18 | 18 | 18 | 72 |
| Area (m$^2$) | 144.72 | 36.00 | 32.94 | 30.60 | 142.56 |
| Volume (m$^3$) | 86.48 | 27 | 19.27 | 16.83 | 96.64 |
| Module No. | 15 | 3 | 3 | 3 | 12 |
|  | Office | Hospital | Hospital | Hospital | Hospital |
| MEP Service | | | Duct, Pipe, Tray | | |

In Table 1, the length represents the length of the corridor to which the PMR is applied, and the length of the module was 6 m considering the ease of pipe assembly, processing, and movement. Accordingly, the number of modules was calculated. The area represents the horizontal area of the corridor to which the PMR was applied, and the volume represents the plenum space corresponding to the area to which the PMR was applied.

The length of the corridor to which Type A was applied was 90 m. The length of the corridor where Type B, C, and D were applied was 18 m. The small length of the corridor may affect productivity, but because production continued in the same place, we judged that the productivity of the workers would be similar. In the case of Type E, the corridor length was 72 m. The length, area, and volume to which each module was applied were measured.

In the production stage, the production was conducted under the guidance and detailed supervision of a general contractor manager, and the investigation was conducted based on the resource inputs reported by each manufacturer.

## 4. Analysis of Economical Structural Framing

### 4.1. Proposed PMR Framing Method

The factor that has the greatest influence on the increase in cost during prefabrication is material cost, and economic structural framing is required to lower the production cost of PMRs [10]. Therefore, this study proposed five PMR structural framing methods, which are as follows:

- Type A: frames are in the form of a continuous set of cubes (supporting interval: 1.25 m).
- Type B: frames are in the form of a continuous set of cubes (supporting interval: 2 m).
- Type C: frames in which the ducts are moved to the outside (supporting interval: 2 m).
- Type D: frames that can be directly fixed to the concrete (supporting interval: 2 m).
- Type E: frames with only the lower support retained (supporting interval: 1.5 m).

Type A is the most commonly used type of frame [10]. In Type A, it is possible to support each member at the bottom and fix a member to the system channel at the top. Because it is in the form of a continuous set of cubes, workers were expected to produce it the fastest in the production stage. It has the most stable form of production, transportation, and construction at the construction site.

In Type B, the supporting interval of the PMR frame was altered from 1.5 m (which was the supporting interval in Type A) to 2 m. By adjusting the supporting interval, the length of system channels required was expected to reduce. To counteract the deterioration of the PMR unit's overall rigidity following the adjustment of the interval, vertical and horizontal cube-shaped frames were added.

In Type C, the duct inside the continuous cube-shaped structural frames was removed and made protruding to the outside. This could reduce the input length of system channels. The ducts occupy more space than the pipe and tray members do; hence, the efficiency of inputting the system channel member was expected to improve when the ducts are removed.

Type D is a structural frame designed to attach PMR modules to concrete. The number of system channels is expected to increase because the members added are cubic. However, it could be applied if it was necessary to respond to the lateral load.

Type E was completely different from the PMR cubic structural frames; only the bottom-supporting member was retained, and only the essential member was added to the top. The number of system channels was expected to be the least for this design.

The five designs and photos (captured by the subcontractor, GS Neotek, Seoul, Republic of Korea) of the structural frames proposed in this study are shown in Figures 1–5.

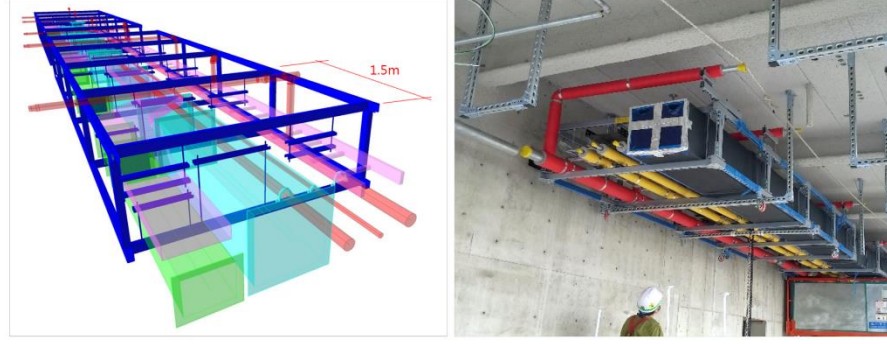

**Figure 1.** Type A: Frames in the form of a continuous set of cubes (supporting interval: 1.25 m).

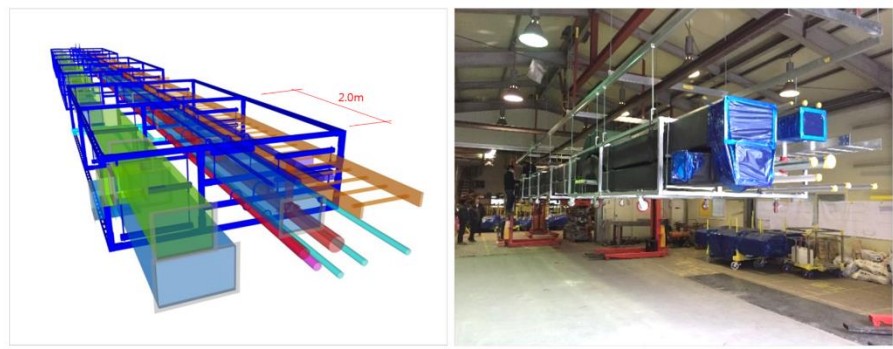

**Figure 2.** Type B: Frames in the form of a continuous set of cubes (supporting interval: 2 m).

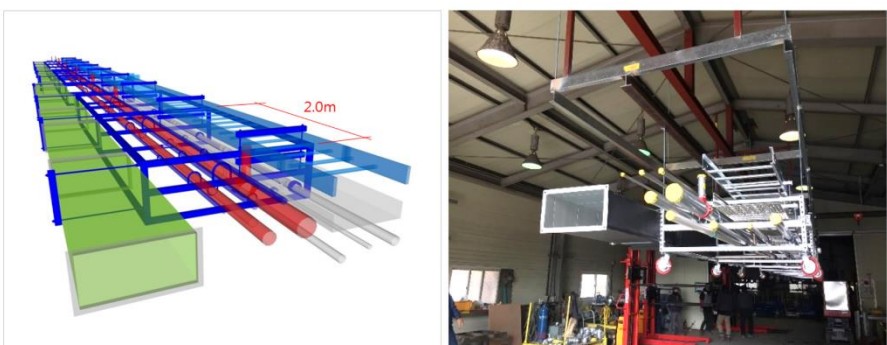

**Figure 3.** Type C: Frames with the ducts moved to the outside (supporting interval: 2 m).

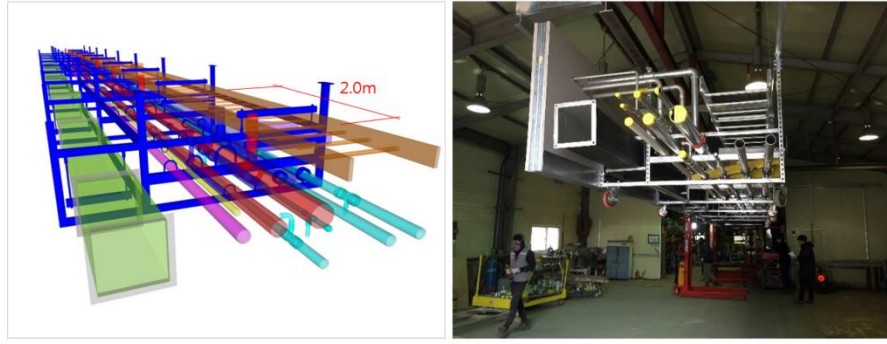

**Figure 4.** Type D: Frames that can be directly fixed to the concrete (supporting interval: 2 m).

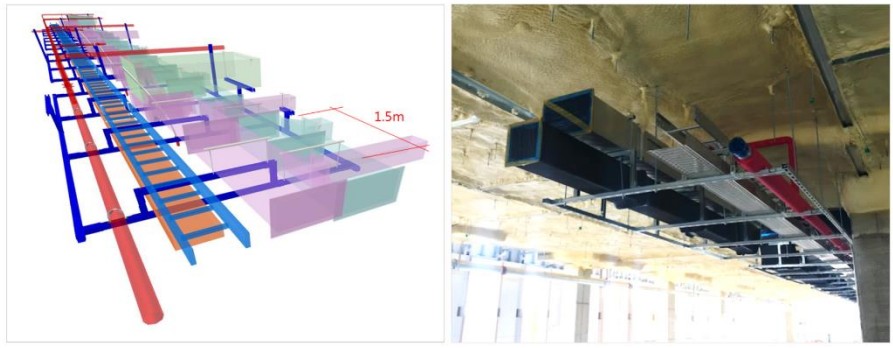

**Figure 5.** Type E: Frames with only the lower support retained (supporting interval: 1.5 m).

### 4.2. Design Cost Analysis

First, the labor input required to design the PMR structural frames was investigated (Figure 6). Because PMRs are mainly applied where the MEP systems are complex [10], BIM was applied, which was expected to increase efficiency, and BIM engineer input was calculated. Based on the primarily coordinated MEP drawings, each BIM engineer calculated the amount of input to the design of the system channel, detail coordination for attachment, and production of spool drawing for the workers' production.

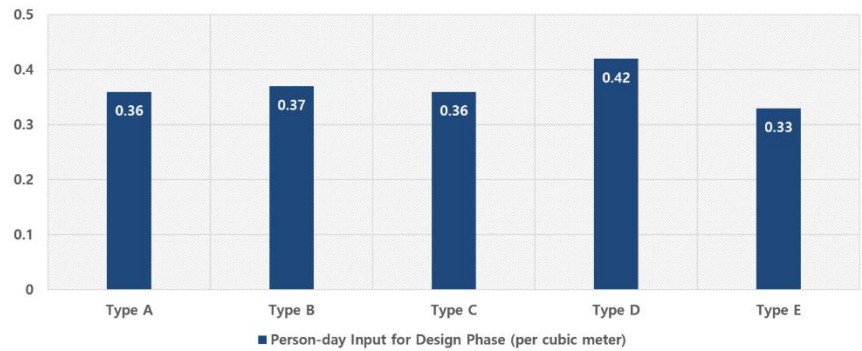

**Figure 6.** Design cost analysis (person-day input for design phase).

Based on the analysis results of the design, a similar level of labor input was required to design Type A, Type B, and Type C structural frames. Person-day BIM engineer inputs for Type A, Type B, and Type C were 0.36, 0.37, and 0.36, respectively, per cubic meter. Type D and Type E were found to require a relatively high level of labor inputs of 0.42 and 0.33, respectively. The latter was the least number of people required.

Type A, Type B, and Type C were all designed with cube-type structural frames. This characteristic was expected to reflect in the BIM engineer input. Consequently, the resource inputs earmarked for the above were the lowest. Although the same design pattern was followed in Type D structural frames, the BIM engineer person-day input for Type D was expected to increase because of the addition of a member for attaching to concrete. Type E frames were not cubic, and the supports to the frames were constructed according to the MEP member. For example, the design was optimized according to whether it was in or out of the MEP system. This was expected to increase the resource inputs; however, in contrast to the expectation, it showed the lowest value. This is attributed to the relatively less design involved for the Type E members.

### 4.3. Additional Material Cost Analysis

Second, the length of the system channel required to construct the PMR structural frames was calculated (Figure 7). The average construction cost per square meter in South Korea from 2015 to 2022 (when the case study was conducted) was approximately 1500 USD (Korea Public Procurement Service), and the unit price of the applied system channel per meter was 15 USD. Even if only one meter of system channel is used for a 1-square-meter rack, the construction cost increases by 1%. The cost of electrical equipment construction alone increases further. Hence, the PMR structural frames significantly impact the total construction cost, and, therefore, this study proposed an economical way of constructing the structural frames. In this study, the input length of the system channel per cubic meter of the plenum space was calculated to analyze the economic feasibility of each PMR-structural-frame design.

In Type A frames, the supporting interval of structural frames was fixed at 1.25 m. Accordingly, the system channel per cubic meter was the highest: 6.75 m. When calculated with respect to a 1-square-meter area, a system channel of length 4 m and an additional 60 USD were required as input. This corresponded to an increase of 4% in the total construction

cost. Type B and Type C were found to be the most economical alternatives in terms of material input because the supporting interval of the structural frame was set to 2 m.

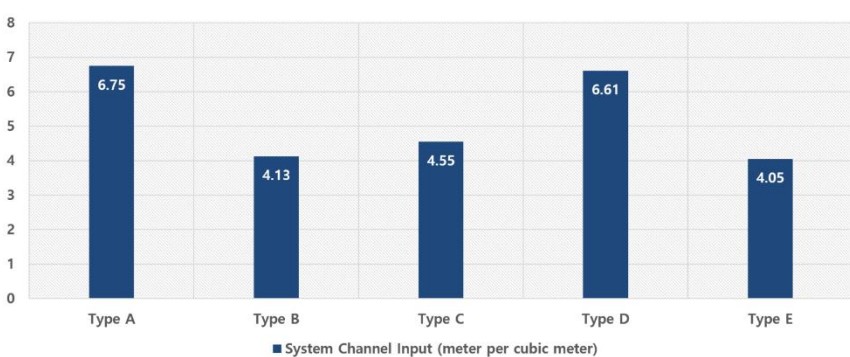

**Figure 7.** Material cost analysis (system channel input for manufacturing).

The input lengths of the system channels required for Type B and Type C were 4.13 m and 4.55 m, respectively, per cubic meter. In Type D, where the frames were attached to the concrete surface of the ceiling, the supporting interval was adjusted to 2 m; however, the input system- channel length was 6.61 m per cubic meter, similar to Type A. In Type E, the supporting interval was set to 1.5 m. This was predicted to be the most economical as the upper system channel layer was removed; however, in terms of the actual material input, the system channel input was similar to that of Type B. Both methods, adjusting the supporting intervals of structural frames (Type B) and designing structural framing only on the necessary areas (Type E), caused a reduction in the material cost.

*4.4. Fabrication Cost Analysis*

Finally, the person-day input required to produce one cubic meter of PMR units was calculated (Figure 8). As of 2022, the unit labor cost for intermediate engineers in Korea, where the case study was conducted, is 175 USD per person-day (Statistics Korea). If one person-day is required per square meter, the construction cost is increased by 12%. Additional labor needed for framing may hinder the application of prefabrication.

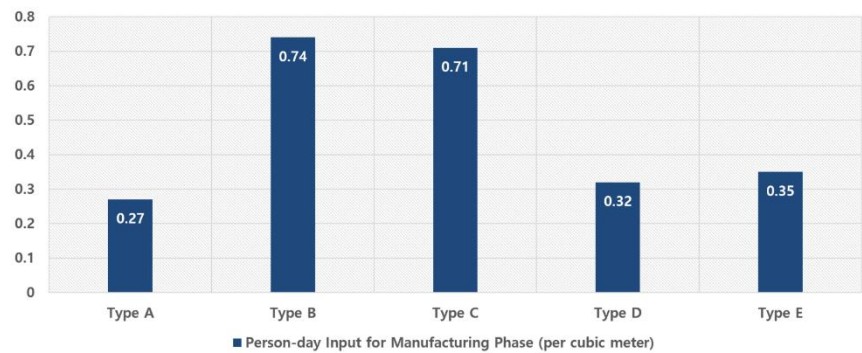

**Figure 8.** Manufacturing cost analysis (person-day input for manufacturing phase).

Type A required a relatively low level of 0.35 person-day per cubic meter. Assuming a total construction cost of 1500 USD, the construction cost increased by 4%. Type B required a person-day input of 0.32, which was slightly lesser than that of Type A. The material input for the system channel of Type B was 61% of that of Type A; however, no significant differences were observed between their labor costs. The decrease in material input alone did not considerably affect the person-day input in the manufacturing stage because of parallel activities, such as the preparation of the lower pedestal of the frames to construct a 1-unit module and matching the horizontality of the modules.

The highest personal-day input was required in the production stage for Type C and Type D. The material input for constructing Type C frames was relatively low; however, a decrease in material input did not lead to a decrease in the person-day input of production. In Type C, structural frames had to be constructed for fixing ducts, and using the basic system channels and both methods simultaneously was expected to be disadvantageous in terms of fabrication effort. In Type D, the most labor cost was invested in the fabrication stage, which is believed to have been affected by the increase in system channel material input.

In terms of fabrication effort, Type E was the most advantageous. The fabrication effort required for Type E was expected to be relatively high because the shape of each multi-trade prefabrication module was slightly changed. However, the analysis results indicated that it required the lowest person-day input: 0.20. Because there was no work on the upper member and most of the work was done on the lower pedestal, the high convenience of fabrication increased the fabrication efficiency.

## 5. Economic Analysis

This section discusses the overall economic feasibility evaluated based on the module design effort, additional material input, and fabrication effort analyzed in the previous section.

The economic feasibility was evaluated using the following:

$$\text{Structural Framing Cost} = \frac{\text{Design Cost}}{\text{m}^3} + \frac{\text{Material Cost}}{\text{m}^3} + \frac{\text{Fabrication Cost}}{\text{m}^3}, \quad (1)$$

where

Design Cost = Engineering Input (person-day) × Labor Cost (USD per day);
Material Cost = System Channel Input (meter) × Unit Cost (USD per day);
Fabrication Cost = Fabrication Effort Labor Input (person-day) × Labor Cost (USD per day).

Because the size of PMR is affected by length, width, and height, the basic unit of economic analysis is area (cubic meters). Structural framing cost is the sum of the design cost (person-day input for design phase), material cost (system-channel input for manufacturing), and fabrication cost (person-day input for manufacturing phase) per cubic meter, as analyzed in Section 4.

The cost of BIM engineers, the cost of the system channel, and the labor cost of system channel assembly engineers were calculated using Korean price information (www.kpi.or.kr, accessed on 11 June 2022) and the input per cubic meter discussed in Section 3, and the total cost per cubic meter was calculated. The calculated results are tabulated in Table 2. The numbers in brackets corresponding to the total design, material, and fabrication costs represent the ratio of each cost to the total structural framing cost.

**Table 2.** Results of economic analysis.

| Phase | | Type A | Type B | Type C | Type D | Type E |
|---|---|---|---|---|---|---|
| Design Cost (per cubic meter) | Engineering Input (person-day) | 0.36 | 0.37 | 0.36 | 0.42 | 0.33 |
| | Labor Cost (USD per day) | 198.57 | 198.57 | 198.57 | 198.57 | 198.57 |
| | Total Design Cost (USD) | 71.49 (29.61%) | 73.47 (37.07%) | 71.49 (25.62%) | 83.40 (25.45%) | 65.53 (38.67%) |
| Material Cost (per cubic meter) | System Channel Input (meter) | 6.75 | 4.13 | 4.55 | 6.61 | 4.05 |
| | Unit Cost (USD per meter) | 15.0 | 15.0 | 15.0 | 15.0 | 15.0 |
| | Total Material Cost (USD) | 101.25 (41.94%) | 61.95 (31.26%) | 68.25 (24.46%) | 99.15 (30.25%) | 60.75 (35.85%) |

**Table 2.** *Cont.*

| Phase | | Type A | Type B | Type C | Type D | Type E |
|---|---|---|---|---|---|---|
| Fabrication Cost (per cubic meter) | Labor Input (person-day) | 0.35 | 0.32 | 0.71 | 0.74 | 0.22 |
| | Labor Cost (USD per day) | 196.2 | 196.2 | 196.2 | 196.2 | 196.2 |
| | Total Fabrication Cost (USD) | 68.67 (28.45%) | 62.784 (31.68%) | 139.302 (49.92%) | 145.188 (44.30%) | 43.164 (25.47%) |
| Total Structural Framing Cost (USD per cubic meter) | | 241.41 | 198.20 | 279.04 | 327.74 | 169.44 |
| Total Structural Framing Cost (USD per square meter) | | 144.26 | 148.65 | 163.24 | 180.26 | 107.41 |

The most economical structural framing was Type E. Type E showed the most economical values in all aspects of design, material, and fabrication costs. Initially, it was predicted that there would be inefficient parts in the design and the fabrication stages; however, the two stages in the case study also showed the lowest cost input. The second-best economical framing was Type B. Type B, with which economic feasibility was pursued by expanding the supporting interval, generally required low design, material, and fabrication costs. Adjusting the supporting interval within a range that ensures the safety of the PMR module may be one method to produce economic structural framing.

The third best economical structural framing was Type A. Type A was found to be the most disadvantageous in terms of material cost; however, its design and fabrication costs were relatively low. The ratio of material cost to total cost was 41.94%, which was the highest. If the material cost in the region where PMR is applied is lower than in other regions, a higher economic feasibility of Type A may be achieved.

The fourth best economical structural framing was Type C. The design and material costs required for Type C frames were relatively low; however, the fabrication cost was high. The ratio of fabrication cost to total cost was 49.92%, the highest among all types. If automated facilities that can increase labor productivity in the fabrication stage are introduced, the economic feasibility can be improved. The least economical framing method was Type D, which required high material and fabrication costs.

## 6. Conclusions

In this study, we analyzed the application of various framing methods to improve the economic feasibility of additional structural framing input, one of the obstacles to multi-trade prefabrication application. We analyzed their design, material, and fabrication costs.

Based on the calculation results of the cost invested in each stage, the cost required only for PMR structural framing was at least 7.2% (Type E) and as high as 12.0% (Type D) of the total construction cost. An analysis was conducted to identify an economical method for structural framing, and the major analysis results are as follows:

1. Design Cost Analysis: the cost incurred by the workforce for the design stage was 25.45–38.67% of the total structural framing cost. Even if the supporting interval was adjusted and the cube shapes repeated, such as in Type A, Type B, and Type C, the effect on the design cost was not significant. In Type E, the lowest cost input was observed despite designing the elements that support the MEP line individually.

2. Material Cost Analysis: 24.46–41.94% of the structural framing cost was spent on PMR framing. The material cost inputs of Type B and Type C, which pursued economic feasibility by increasing the supporting interval, were lower than that of others. Regarding material cost, Type E, in which the upper framing was removed and only the parts that require MEP supporting were designed, was the most economical.

3. Fabrication Cost Analysis: fabrication cost accounted for 25.47–49.92% of the total structural framing cost. Adjusting the supporting interval reduced the fabrication cost but did not generate a significant difference in the ratio of material quantity. This



indicates that fabrication cost is not directly affected by quantity. Type C was identified to be disadvantageous in terms of fabrication cost. This suggests that simplifying the fabrication pattern may be more efficient than reducing the quantity. The most economical framing scheme was Type E, with decreased overall material quantity.

The contributions of this finding to academia and industry are as follows. Existing studies have only compared the economics of PMR application or suggested additional effects that can be obtained from PMR application. In addition, some studies report that using PMR increases costs, whereas others report that it decreases costs.

Economic feasibility can be improved through detailed changes in construction methods and the development of details. In PMR, structural framing incurs more than 7.2% of the total construction cost, and this study provided a systematic analysis that could improve the economic feasibility of structural framing. The analysis was conducted in terms of design, material, and fabrication costs. Another economic type of structural framing is possible by reflecting regional characteristics.

This study inherited the limitations of case studies. It has a small sample size and aspects that are difficult to generalize. However, detailed PMR design options were presented, unlike the results reported in previous studies, and the effect of applying them was analyzed in detail. Through the analysis of the design stage and the manufacturing process in the factory, the input of labor was measured by time, and the input of materials was calculated. Although not all conditions can be similar to the environment in which this study was conducted, the findings can help enhance the economic feasibility of prefabrication and modularization of construction and their widespread utilization.

**Author Contributions:** Conceptualization, S.B., J.W. and S.J.; methodology, S.B. and S.J.; validation, S.B. and J.W.; formal analysis, J.W. and S.J.; investigation, S.B.; resources, J.W. and S.J.; data curation, J.W. and S.J.; writing—original draft preparation, S.B.; writing—review and editing, J.W. and S.J.; visualization, S.B.; supervision, J.W. and S.J.; project administration, J.W. and S.J.; funding acquisition, S.J. All authors have read and agreed to the published version of the manuscript.

**Funding:** This research was supported by Basic Science Research Program through the National Research Foundation of Korea (NRF), funded by the Ministry of Education [2022R1C1C1005963].

**Institutional Review Board Statement:** Not applicable.

**Informed Consent Statement:** Not applicable.

**Data Availability Statement:** Not applicable.

**Conflicts of Interest:** The authors declare no conflict of interest.

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
