# Peer review of "Economic Integrated Structural Framing for BIM-Based Prefabricated Mechanical, Electrical, and Plumbing Racks"

_applsci, doi:10.3390/app13063677_

Round 1
Reviewer 1 Report
The paper is interesting and deserves publication. However, the reviewer has some significant concerns regarding the paper methodology. The authors totally neglect the bearing capacity of the racking system. The reviewer believes that it should be somehow included in the assessment. If the bearing capacity is higher, more services can be guaranteed and fewer racks will be needed. Which is the benefit of using a single racking system with higher capacity or two with lower capacity? This is an important aspect that should be included in the economic assessment. It is not possible to present an evaluation neglecting the design requirements.
The reviewer is willing to see the revised version of this paper.
Author Response
In response to the recommendation of the reviewer, the authors clarified the research methods and detailed the design requirements.

Reviewer 2 Report
The authors carry out research for an economic analysis for prefabricated mechanical, electrical, and plumbing (MEP) rack (PMR) structural framing.
The study proposed by the authors can contribute to improving the economic feasibility of prefabrication and modularization of construction and their widespread use.
Further clarification is needed to justify why the study proposed five methods of structural framing of PMR
In the Research Method section, a case study is used for Table 1. The authors did not detail how the values ​​in the table were defined or their source.
The authors must add the bibliographic source for the presented Figures (all).
Provide must detail for the way in which they chose to define the formula for Structural Framing Cost.
The section is not completed - Author Contributions. Also, section Data Availability Statement: must be completed or deleted
Author Response
The authors would like to thank the reviewers for their careful and constructive comments. Detailed responses to the specific points raised are given in attached response letter.

Round 2
Reviewer 1 Report
The paper can now be considered for publication.
Reviewer 2 Report
In this form of the manuscript, the authors have implemented those suggested by the reviewers.